# FINE-TUNING CAN CRIPPLE FOUNDATION MODELS; PRESERVING FEATURES MAY BE THE SOLUTION

## ABSTRACT

Pre-trained foundation models, due to their enormous capacity and their training using vast amounts of data can store knowledge about many real-world concepts. To further improve performance on downstream tasks, these models can be fine-tuned on task specific datasets. While various fine-tuning methods have been devised and have been shown to be highly effective, we observe that a fine-tuned model's ability to recognize concepts on tasks *different* from the downstream one is reduced significantly compared to its pre-trained counterpart. This is clearly undesirable as a huge amount of time and money went into learning those very concepts in the first place. We call this undesirable phenomenon "concept forgetting" and via experiments show that most end-to-end fine-tuning approaches suffer heavily from this side effect. To this end, we also propose a rather simple fix to this problem by designing a method called LDIFS (short for $\ell_2$ distance in feature space) that simply preserves the features of the original foundation model during fine-tuning. We show that LDIFS significantly reduces concept forgetting without having noticeable impact on the downstream task performance.

## 1 INTRODUCTION

Foundation models like CLIP Radford et al. (2021), ALIGN Jia et al. (2021) and CoCa Yu et al. (2022) are trained using self-supervised methods on hundreds of millions or even billions of samples scraped from the internet. This massive, compute intensive pre-training makes such models a knowledge store on a vast number of real-world concepts, enabling them to easily transfer to a wide variety of downstream tasks and applications. Indeed, this ability to recognize real-world concepts and thereby transfer to downstream tasks is the primary advantage of such models and is the very reason behind their name Bommasani et al. (2021).

While a foundation model can achieve impressive performance on a downstream task often without even requiring a single training sample from the task itself Radford et al. (2021), in order to maximise performance, it conventionally requires some form of fine-tuning on the task at hand. There are multiple types of fine-tuning methods like linear probing Radford et al. (2021), prompt-tuning Zhou et al. (2022b;a), adapters Gao et al. (2021); Zhang et al. (2021), weight-space interpolation Wortsman et al. (2022b); Ilharco et al. (2022a) and full end-to-end fine-tuning Radford et al. (2021); Kumar et al. (2022); Goyal et al. (2022); Xuhong et al. (2018). Among these types, full end-to-end fine-tuning is well-known to produce the best performance on downstream tasks.

It is worth noting here that the pre-training dataset of a foundation model, owing to its massive scale, contains information about several thousands of real-world concepts. Hence, it is highly likely that the downstream dataset for fine-tuning the model will only contain a significantly smaller number of concepts compared to its pre-training set. A natural question that arises then is: *How does end-to-end fine-tuning of a foundation model affect the vast knowledge it acquired through its pre-training?* In this work, this is precisely what we aim to answer.

Through a thorough study of popular end-to-end fine-tuning methods, we observe that for most of them, the fine-tuned model has significantly lost its ability to recognize real-world concepts outside the downstream task, a phenomenon which we call *concept forgetting*. This is a highly undesirable effect as there are important use-cases where we would want the foundation model's vast knowledge on concepts to be preserved after fine-tuning.

For instance, one such use-case is the requirement for *continual fine-tuning* to incorporate previously unknown information into a pre-trained model. Companies like OpenAI and Google have spent millions of dollars pre-training large foundation models from scratch. Even so, these powerful models do tend to have gaps in their knowledge. Specific examples are ChatGPT OpenAI (2020) not knowing about events after September 2021 or CLIP failing to classify satellite images or detect lymph node tumors Radford et al. (2021). Naturally, retraining the foundation model from scratch by combining the new data with prior training data, is too expensive to be a feasible option. Such scenarios require the model to be fine-tuned only on the new data while preserving its prior knowledge at the same time, thereby necessitating investigation into fine-tuning methods which prevent concept forgetting.

Although conventional end-to-end fine-tuning methods generally suffer from concept forgetting, we find that there can be a relatively simple fix to this problem. In particular, we observe that if a fine-tuning method ensures that the fine-tuned model is close in some sense to the original foundation model, it can significantly reduce concept forgetting. While the vicinity of the original model can be defined in terms of distance in the parameter space, as seen in the case of the L2SP regularizer Xuhong et al. (2018), we find that it is much more effective to define it in terms of distance in the model's feature space which captures its true input-output behaviour. This leads us to propose a new regularizer, **LDIFS ($\ell_2$ distance in Feature Space)**, which minimizes the distance between the features of the original model and those obtained from the model being fine-tuned. Moreover, we observe that simply preserving the last-layer features is not effective in reducing concept forgetting. Motivated from observations in Zhang et al. (2018), we preserve features extracted from different internal representations. We find that this relatively simple method of preserving the original model's features while fine-tuning on a downstream task, can significantly alleviate the problem of concept forgetting in the fine-tuned model, without affecting its performance on the downstream task. Thus, to summarize, our contributions in this work are:

1. **Investigate concept forgetting.** We perform a thorough evaluation, benchmarking 6 existing end-to-end fine-tuning methods on 10 different downstream tasks and find that concept forgetting, as a phenomenon, exists in all of them.
2. **Analyze different end-to-end fine-tuning methods.** We find a consistent ordering in the ability of different fine-tuning methods to avoid concept forgetting. Particularly, we find the L2SP Xuhong et al. (2018) regularizer to outperform other fine-tuning baselines. We analyze why this is the case.
3. **Propose a new regularizer for end-to-end fine-tuning.** Analyzing L2SP helps us propose a simple new regularizer (LDIFS) which minimizes feature space distance while fine-tuning.
4. **A nudge towards continual fine-tuning.** Finally, as a natural extension of the evaluation setting we evaluate on a sequence of multiple fine-tuning tasks and find LDIFS to be superior to other fine-tuning methods even in this setting.

## 2 A BRIEF NOTE ON FINE-TUNING

Here we provide a description of CLIP Radford et al. (2021) as our foundation model of choice and briefly discuss existing state-of-the-art methods used for fine-tuning CLIP.

**CLIP, a brief overview** Broadly speaking, a CLIP model has two components: **i)** the vision or image encoder[1] $f_{\theta_v} : \mathbb{R}^{C,H,W} \to \mathbb{R}^D$, and **ii)** the text encoder $f_{\theta_t} : \mathbb{R}^L \to \mathbb{R}^D$. CLIP is pre-trained on 400 million pairs of images and corresponding text descriptions scraped from the internet. For pre-training, it uses a contrastive loss to maximize cosine similarity between the correct (image, text) pairs and minimize the same for the incorrect ones. Due to its large-scale self-supervised pre-training, CLIP exhibits impressive performance on several downstream tasks, often without requiring a single training sample from the task itself. However, in order to maximise performance on a specific downstream task, the pre-trained CLIP model is conventionally fine-tuned further on the task itself. Below we provide a brief description of popular such fine-tuning methods relevant to our study.

**Zero-shot (ZS)** In image classification, given an input image $\mathbf{x}$ and a set of $K$ class names $\{\mathbf{c}_i\}_{i=1}^K$ as natural language text, the $D$-dimensional encoding[2] for each class name $\psi(\mathbf{c}_i) = f_{\theta_t}(\mathbf{c}_i)$ and the image $\phi(\mathbf{x}) = f_{\theta_v}(\mathbf{x})$ are first obtained. The text encodings $\psi(\mathbf{c}_i)$ are then used as parameters of a

---

[1]Note, the vision encoder first extracts $D_v$ dimensional image features and then projects them to a $D$ dimensional space via a linear embedder $\mathbf{w}_v : \mathbb{R}^{D_v} \to \mathbb{R}^D$. Similarly for the text encoder.

[2]The natural language class names are often augmented with a prompt like "an image of a {class name}."

$K$-class linear classifier, and the classification inference on $\mathbf{x}$ is performed as $\arg\max_i \psi(\mathbf{c}_i)^{\mathrm{T}}\phi(\mathbf{x})$. This is known as the zero-shot (ZS) prediction and CLIP's best model has been shown to have competitive ZS accuracies with a fully supervised ResNet-101 on ImageNet Radford et al. (2021).

**Linear Probe (LP)** In this case, an additional linear layer $\mathbf{w} \in \mathbb{R}^{D \times K}$ is appended on top of the image encoder $f_{\theta_v}$ and the weights of this linear layer are trained by solving a standard logistic regression problem (e.g., scikit-learn's `LogisticRegression` module Pedregosa et al. (2011)). The linear layer $\mathbf{w}$ is normally initialized using the text representations $\{\psi(\mathbf{c}_i)\}_{i=1}^{K}$, known as ZS initialization. It is trivial to note that in the absence of any training, LP boils down to ZS.

**End-to-end fine-tuning** While a pre-trained CLIP encoder can obtain impressive ZS and LP accuracies on several tasks, in order to maximize performance on a specific downstream task, the general rule of thumb is to initialize a model from the weights of the pre-trained encoder and then fine-tune the model end-to-end on the downstream task. Here we list some of the most popular end-to-end fine-tuning methods which we study in this work. We provide a more detailed discussion on the different types of fine-tuning methods in §6.

1. **ZS-init-CE** Radford et al. (2021): This is the classic end-to-end fine-tuning method where, similar to the LP, a ZS initialized linear head $\mathbf{w} : \mathbb{R}^D \to \mathbb{R}^K$ is appended to the image encoder $f_{\theta_v}$. However, differently from the LP, parameters of the entire model $\theta = \{\theta_v, \mathbf{w}\}$ (including the image encoder parameters) are fine-tuned using a cross-entropy loss $\mathcal{L}_{\mathrm{CE}}$.

2. **LP-init-CE (LP-FT)** Kumar et al. (2022): This is similar to ZS-init-CE but instead of initializing the appended linear head via ZS, it is initialized by performing linear probing on the downstream task first. Once the linear head is initialized, the entire model is end-to-end fine-tuned using $\mathcal{L}_{\mathrm{CE}}$.

3. **ZS-init-L2SP** Xuhong et al. (2018): In addition to the cross-entropy loss $\mathcal{L}_{\mathrm{CE}}$, this method uses an additional regularizer to minimize the $\ell_2$ distance between the pre-trained and fine-tuned *image encoder weights*, thereby tries to keep the fine-tuned model weights close to the pre-trained ones. Let the pre-trained image encoder weights be $\theta_{v(0)}$ and the encoder weights at time step $t$ during fine-tuning be $\theta_{v(t)}$. Then, the fine-tuning loss in this case becomes

$$\mathcal{L}_{\mathrm{L2SP}} = \mathcal{L}_{\mathrm{CE}} + \lambda_{\mathrm{L2SP}}||\theta_{v(t)} - \theta_{v(0)}||_2^2. \tag{1}$$

   Note that the $\ell_2$ distance is only computed between the weights of the pre-trained and fine-tuned image encoders.

4. **LP-init-L2SP**: This is similar to ZS-init-L2SP but the linear head $\mathbf{w}$ is initialized by performing linear probing on the downstream dataset first. The loss for end-to-end fine-tuning then is the same as in Equation (1) [3].

5. **FLYP** Goyal et al. (2022): The Fine-tune Like You Pre-train or FLYP baseline fine-tunes both the image and the text encoders of CLIP and uses contrastive loss $\mathcal{L}_{\mathrm{cont}}$ instead of cross-entropy $\mathcal{L}_{\mathrm{CE}}$ for fine-tuning on the downstream task. The parameters being fine-tuned here are $\theta = \{\theta_v, \theta_t\}$, i.e., both image and text encoders of CLIP.

6. **FLYP-CE** Goyal et al. (2022): This is an ablation on FLYP where, instead of using contrastive loss, the fine-tuning is done using cross-entropy loss $\mathcal{L}_{\mathrm{CE}}$, taking the cosine similarities between image and text embeddings as logits. Note that similar to FLYP, in this case as well, both image and text encoders are fine-tuned end-to-end.

## 3 THE CRIPPLING EFFECT OF END-TO-END FINE-TUNING

The contrastive pre-training dataset of CLIP contains 400 million (image, text) pairs scraped from the internet Radford et al. (2021). Consequently, any downstream task that CLIP is fine-tuned on is highly likely to contain only a small fraction of concepts compared to what it has already been exposed to during pre-training. To investigate the impact of fine-tuning, here we perform a thorough study benchmarking 6 fine-tuning methods on 9 downstream classification tasks. We find that for most of these methods, while the fine-tuned model attains excellent improved performance on the downstream task itself, its general ability to recognize concepts outside the task is significantly reduced over the course of fine-tuning. We call this phenomenon ***concept forgetting*** and find this to be an undesirable effect of most fine-tuning methods. To explore this in more detail, in the remainder of the section, we first discuss how we quantify concept forgetting, then we propose our benchmarking setup for fine-tuning methods and finally present our observations.

---

[3]To the best of our knowledge, LP-init-L2SP, has not been evaluated or benchmarked prior to this work.

### 3.1 Quantifying Concept Forgetting

During ZS and LP evaluation of a model (refer §2) the pre-trained image encoder weights $\theta_v$ remain frozen and unchanged irrespective of the downstream task at hand. Therefore, ZS and LP performance on a specific downstream task can be a good indicator of the pre-trained model's encoded knowledge about the task. This is the hypothesis we base our analysis on.

Furthermore, while ZS accuracy is based on how well the text encoder representations can separate task-specific categories in the image encoder's feature space, LP performance is indicative of whether image encoder representations are linearly separable. This distinction is important as fine-tuning the weights of just the image encoder $\theta_v$ can lead to a situation where its representations are no longer well-aligned with the text encoder. Even so, for a given task, if the image encoder representations are linearly separable, it shows that the model is still able to recognize concepts involved in the downstream task, thereby indicating the pre-trained model's encoded knowledge on the task.

Therefore, in order to quantify concept forgetting on a particular task represented through a dataset $\mathcal{D}$, we simply measure the difference in both ZS and LP accuracy between the pre-trained and fine-tuned image encoders on $\mathcal{D}$. To formalize this, let $f_{\theta_{v(0)}}$ and $f_{\theta_v}$ be the pre-trained and fine-tuned image encoders respectively, and let $\mathcal{D}_{\text{ft}}$ represent the dataset on which we fine-tune $\theta_{v(0)}$. Furthermore, let $\mathcal{A}_{\text{ZS}}(f_{\theta_v}, \mathcal{D})$ and $\mathcal{A}_{\text{LP}}(f_{\theta_v}, \mathcal{D})$ represent the ZS and LP accuracy of image encoder $f_{\theta_v}$ on dataset $\mathcal{D}$. Then, we can define concept forgetting on $\mathcal{D}$ as the change in LP (or ZS) accuracy on $\mathcal{D}$ between pre-trained and fine-tuned models: $\Delta_{\text{LP}}(\mathcal{D}, f_{\theta_v}, f_{\theta_{v(0)}})$ (or in short, $\Delta_{\text{LP}}$) as:

$$\Delta_{\text{LP}}(\mathcal{D}, f_{\theta_v}, f_{\theta_{v(0)}}) = \mathcal{A}_{\text{LP}}(f_{\theta_v}, \mathcal{D}) - \mathcal{A}_{\text{LP}}(f_{\theta_{v(0)}}, \mathcal{D}) \tag{2}$$

We can define $\Delta_{\text{ZS}}(\mathcal{D}, f_{\theta_v}, f_{\theta_{v(0)}})$ in a similar manner. Clearly, as the image encoder $\theta_v$ is obtained after fine-tuning on $\mathcal{D}_{\text{ft}}$, we can expect $\Delta_{\text{LP}}(\mathcal{D}_{\text{ft}}, f_{\theta_v}, f_{\theta_{v(0)}})$ and $\Delta_{\text{ZS}}(\mathcal{D}_{\text{ft}}, f_{\theta_v}, f_{\theta_{v(0)}})$ to increase over the course of fine-tuning. However, the more interesting case is to capture concept forgetting of the model over datasets other than the fine-tuning dataset itself, i.e., $\Delta_{\text{ZS}}(\mathcal{D}, f_{\theta_v}, f_{\theta_{v(0)}})$ and $\Delta_{\text{LP}}(\mathcal{D}, f_{\theta_v}, f_{\theta_{v(0)}})$ when $\mathcal{D} \neq \mathcal{D}_{\text{ft}}$. A negative value for these metrics indicates concept forgetting and zero indicates knowledge accumulation. However, a positive value indicates knowledge gain or a positive forward transfer on the task under inspection.

**Catastrophic forgetting vs concept forgetting**  Catastrophic forgetting McCloskey & Cohen (1989); Kemker et al. (2018); Kirkpatrick et al. (2017) is a well-known phenomenon which signifies how, when a model is trained on a new task, its performance on the previously task drops catastrophically. For example, a model trained on ImageNet is then trained on say CIFAR100, it almost completely loses its performance on ImageNet. Though *concept forgetting* mentioned in this work is very similar to catastrophic forgetting and we do not claim much conceptual novelty here, there still are subtle differences that we believe requires a distinction between the two. The major difference being the fact that in the pre-foundation model era it was known exactly on which task or dataset a model was trained (for example, ImageNet in the example above) and therefore, it was possible to roughly quantify the degree of damage the model had once it was fine-tuned on a new downstream task. However, in the case of foundation models, it is not possible to quantify exactly what the model knows as the model was pre-trained on millions (or even billions) of examples, therefore, it is not possible to quantify exactly what the fine-tuned model forgot and the catastrophe therein. Hence, it is necessary to devise poking mechanisms similar to the ones presented in this work using ZS and LP, to quantify the effect (forgetting) of fine-tuning on a small but useful domain of concepts (represented by the dataset $\mathcal{D}$ used to quantify $\Delta_{\text{LP}}$ or similar metrics).

### 3.2 Benchmarking concept forgetting

To quantify concept forgetting, here we use CLIP Radford et al. (2021) ViT-B/32 pre-trained on the OpenAI dataset and released in the OpenCLIP repository Ilharco et al. (2021) and measure its ZS and LP performance over fine-tuning on 10 different image classification downstream tasks with a high variability in their semantic concepts. These datasets, along with their respective train/test splits are: **a)** *Stanford Cars* Krause et al. (2013) containing 16185 images of 196 classes of cars with a train/test split of 8144 and 8041 images respectively, **b)** *CIFAR-10/100 (C10/100)* Krizhevsky et al. (2009) containing 60000 images of vehicles, flora and fauna, divided into 10/100 classes with the train/test split for both C10 and C100 having 50000 and 10000 images respectively, **c)** *DTD*

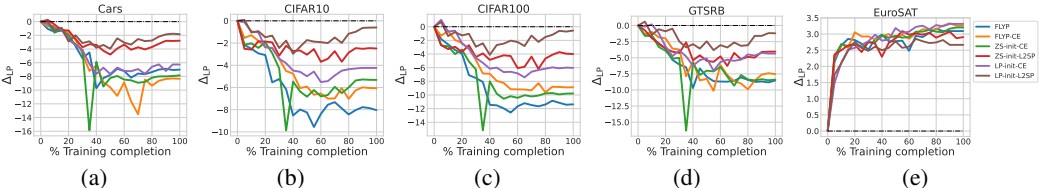

Figure 1: **Test set $\Delta_{\mathrm{LP}}\%$ for models fine-tuned on EuroSAT using different fine-tuning methods.** While EuroSAT $\Delta_{\mathrm{LP}}$ rises, $\Delta_{\mathrm{LP}}$ on all other datasets is almost always negative throughout the fine-tuning with a sole exception of DTD when fine-tuned using LP-init-L2SP. See §3.2 for details.

Figure 2: **Test set $\Delta_{\mathrm{LP}}\%$ evaluated on different datasets for models fine-tuned on EuroSAT via various fine-tuning methods.** The L2SP baselines (particularly LP-init-L2SP) have the lowest negative $\Delta_{\mathrm{LP}}$ on datasets other than EuroSAT.

Cimpoi et al. (2014) containing 3760 images of 47 classes of textures found in the wild with 1880 images each in the train and test sets, **d**) *EuroSAT* Helber et al. (2019) containing 25000 samples with 10 categories of satellite images of landscapes and 19600/5400 training/test images respectively, **e**) *GTSRB* Stallkamp et al. (2012) containing 39270 images of 43 classes of German traffic signs with 26640 training images and 12630 test images, **f**) *MNIST* LeCun et al. (1998) containing 60000 training images and 10000 test images of 10 digits from 0 to 9 in grayscale, **g**) *RESISC45(R45)* Cheng et al. (2017) containing 25200 samples with 45 classes of various remote sensing image scenes with the train/test split having 18900 and 6300 images respectively, **h**) *SVHN* Netzer et al. (2011) containing a total of 99289 colour images of street view house numbers, each image being categorized into one of 10 digits with 73257 training samples and 26032 test samples and **i**) *ImageNet* Deng et al. (2009) containing a total of 1.28 million training images and 50000 validation images of 1000 classes. Finally, for this study, we use the 6 end-to-end fine-tuning methods discussed in §2. The training details can be found in Appendix A.

**Fine-tuning causes concept forgetting.** In Figure 1, we present the $\Delta_{\mathrm{LP}}\%$ for models fine-tuned on EuroSAT using the 6 fine-tuning methods discussed in §2. Across all fine-tuning methods, we observe that *while performance on EuroSAT test set increases and $\Delta_{\mathrm{LP}}$ is positive (also see Figure 2e), performance on all 8 other downstream datasets decreases and $\Delta_{\mathrm{LP}}$ for all of them is negative.* This indicates, that *all 6 fine-tuning methods suffer from concept forgetting.* The only exception to this is when we evaluate on DTD and the model is fine-tuned using LP-init-L2SP. We observe $\Delta_{\mathrm{LP}}$ to be mostly positive before it goes to zero at the end of fine-tuning. This is an interesting case as it indicates that LP-init-L2SP on EuroSAT might actually be helping increase knowledge about DTD before the fine-tuning becomes too specific to EuroSAT. This may be an example of positive forward transfer Lopez-Paz & Ranzato (2017); Chaudhry et al. (2018) and exploring why such knowledge accumulation happens is an interesting avenue for further research.

Next, we compare between these fine-tuning methods by showing $\Delta_{\mathrm{LP}}\%$ for these methods for different downstream datasets in Figure 2. Interestingly, *LP-init-L2SP consistently outperforms others in preserving the fine-tuned model's original performance across multiple downstream tasks*, as is observable from its low negative $\Delta_{\mathrm{LP}}$ compared to other baselines. While it suffers an initial dip in performance in the early stages of fine-tuning, in the later stages, LP-init-L2SP regains the accuracy, often ending up with a near zero $\Delta_{\mathrm{LP}}$. Its impressive performance however, does seem to come at the cost of a relatively lower positive $\Delta_{\mathrm{LP}}$ on the fine-tuning task, i.e., EuroSAT, itself. Our observations for other datasets are similar (see Appendix C). In what follows, we first investigate LP-init-L2SP further to gain some insights on why it preserves concepts better than the other baselines, and then use those insights to propose a new fine-tuning method that significantly outperforms all other baselines.

## 4  CAN PRESERVING FEATURES HELP?

The L2SP regularizer (eq. (1)) enforces the model $f_{\theta_{v(t)}}$ at time-step $t$ of fine-tuning to be in the vicinity of the pre-trained model $f_{\theta_{v(0)}}$ by minimizing the $\ell_2$ distance between the two in the

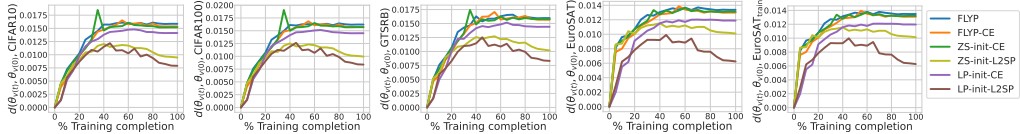

Figure 3: $\ell_2$ **distance in the parameter space** $||\theta_{v(t)} - \theta_{v(0)}||_2^2$ of the image encoder over the course of fine-tuning. Except L2SP, all other baselines diverge away from their pre-trained counterparts.

Figure 4: $\ell_2$ **distance in feature space** $d(\theta_{v(t)}, \theta_{v(0)}, \mathcal{D})$ between image encoders computed over the course of fine-tuning on EuroSAT.

parameter space. As evident from §3, this simple regularizer has a promising impact on the model's ability to avoid concept forgetting. To understand how correlated the parameter space $\ell_2$ distance $||\theta_{v(t)} - \theta_{v(0)}||_2^2$ is to concept forgetting, in Figure 3, we show how $\ell_2$ distance changes as we fine-tune different datasets (EuroSAT, GTSRB and SVHN) using all the discussed fine-tuning methods. The observation is crystal clear. *Relatively speaking, all the fine-tuning methods except the two L2SP baselines (ZS-init-L2SP and LP-init-L2SP) cause the parameters of the fine-tuned model to move away from its pre-trained counterpart.* For the L2SP baselines, $||\theta_{v(t)} - \theta_{v(0)}||_2^2$ first increases and then falls back down indicating that the model first diverges slightly and then gets closer again to the pre-trained model in the parameter space.

Though keeping the fine-tuned model in the vicinity of the pre-trained one shows a promising effect in preserving concepts, however, vicinity in the parameter space need not necessarily capture the true input-output behaviour of the pre-trained model. In fact, it is trivial to construct two different sets of model parameters, far in $\ell_2$ distance, outputting similar values in a specific domain. Additionally, regularizing in the parameter space might not keep the fine-tuned model in the *desired* vicinity that preserves the pre-trained knowledge and performs well on the downstream task at the same time. Therefore, we conjecture that vicinal distance in the feature space might be a better indicator of how effectively models can preserve concepts. Indeed, in the end, the internal representation space is where models encode patterns. In Figure 4, we inspect $\ell_2$ distance in feature space over fine-tuning using the following distance function:

$$d(\theta_{v(t)}, \theta_{v(0)}, \mathcal{D}) = \frac{1}{N} \sum_{i=1}^{N} ||\Phi_{\theta_{v(t)}}(\mathbf{x}_i) - \Phi_{\theta_{v(0)}}(\mathbf{x}_i)||_2^2, \tag{3}$$

where, $\Phi_{\theta_{v(t)}}(\mathbf{x}_i)$ represents the features of the model with parameters $\theta_{v(t)}$ at time $t$ for a given sample $\mathbf{x}_i$, and $N$ is the number of samples in the dataset. Note that the feature vector $\Phi_{\theta_{v(t)}}(\mathbf{x}_i)$ is obtained by concatenating various internal representations (not just the last layer features) of the network architecture, similar to the perceptual features presented in Zhang et al. (2018). Exact details as to how we compute the concatenated feature vector for a ViT-B/32 model is mentioned in Appendix B, and similar plots for other datasets is shown in Appendix C.

**A strong correlation between concept forgetting and feature-space distance.** Similar to our observations in Figure 3 (parameter space), we note that except L2SP, all other fine-tuning methods suffering significantly from concept forgetting cause the fine-tuned model to move away from the pre-trained model in terms of feature-space distance as well (Equation (3)). In case of ZS-init-L2SP and LP-init-L2SP, while initially diverging, the fine-tuned models recover their pre-trained behaviour to a certain extent in the later stages of fine-tuning. It is important to note that this observation is consistent for the fine-tuning training and test sets as well as for all other datasets.

Based on the empirical results shown in Figure 2 and Figure 4 we conclude that the *fine-tuning methods which cause the model to significantly diverge away from the pre-trained model either in parameter space or in feature space suffer more from concept forgetting*. This observation, combined with our conjecture leads to a natural extension of fine-tuning where we use $d(\theta_{v(t)}, \theta_{v(0)}, \mathcal{D})$ (distance in the feature space) as the regularizer. We call this regularizer **LDIFS:** $\ell_2$ **distance in Feature Space**. The

Figure 5: **Test set $\Delta_{\text{LP}}\%$ evaluated on different datasets for models fine-tuned on EuroSAT.** LDIFS (Our) baselines (both ZS-init and LP-init) provides the best results in terms of avoiding concept forgetting without affecting downstream performance on EuroSAT.

| Dataset | | Fine-tuning baselines | | | | | | | | | | | | |
| Fine-tune | Eval | FLYP | | FLYP-CE | | ZS-init-CE | | LP-init-CE (LP-FT) | | ZS-init-L2SP | | LP-init-L2SP | | ZS-init-LDIFS (Ours) | | LP-init-LDIFS (Ours) | |
| | | $\Delta_{\text{ZS}}(\uparrow)$ | $\Delta_{\text{LP}}(\uparrow)$ | $\Delta_{\text{ZS}}(\uparrow)$ | $\Delta_{\text{LP}}(\uparrow)$ | $\Delta_{\text{ZS}}(\uparrow)$ | $\Delta_{\text{LP}}(\uparrow)$ | $\Delta_{\text{ZS}}(\uparrow)$ | $\Delta_{\text{LP}}(\uparrow)$ | $\Delta_{\text{ZS}}(\uparrow)$ | $\Delta_{\text{LP}}(\uparrow)$ | $\Delta_{\text{ZS}}(\uparrow)$ | $\Delta_{\text{LP}}(\uparrow)$ | $\Delta_{\text{ZS}}(\uparrow)$ | $\Delta_{\text{LP}}(\uparrow)$ | $\Delta_{\text{ZS}}(\uparrow)$ | $\Delta_{\text{LP}}(\uparrow)$ |
|---|---|---|---|---|---|---|---|---|---|---|---|---|---|---|---|---|---|
| Cars | Others | −4.40 | −0.01 | −5.14 | −1.67 | −4.78 | −1.56 | −3.73 | −0.63 | −1.5 | −1.07 | 0.25 | 0.47 | −0.58 | −0.36 | −0.17 | −0.18 |
| | Cars | 9.2 | **5.25** | 5.73 | 3.95 | 18.24 | 2.67 | −5.62 | 4.14 | 9.44 | 1.83 | −3.31 | 3.06 | **20.52** | 4.71 | −3.15 | 4.45 |
| CIFAR10 | Others | −18.79 | −3.35 | −12.85 | −1.17 | −13.95 | −1.6 | −10.41 | −0.81 | −0.82 | 1.04 | 0.99 | 1.16 | 0.06 | 1.03 | **2.03** | **1.18** |
| | CIFAR10 | 7.87 | 2.79 | 7.8 | 2.65 | **8.25** | **2.83** | 7.5 | 2.79 | 8.05 | 2.8 | 6.38 | 2.74 | 7.72 | 2.46 | 6.39 | 2.33 |
| CIFAR100 | Others | −12.89 | −1.16 | −6.54 | −0.5 | −8.63 | −0.96 | −4.6 | −0.11 | −0.33 | 0.65 | 0.33 | 1.03 | −1.05 | 0.82 | −0.3 | 0.86 |
| | CIFAR100 | 24.4 | 9.34 | 23.16 | 9.12 | **24.63** | 8.97 | 17.26 | 8.78 | 23.33 | 8.2 | 13.77 | 7.31 | 24.24 | 8.31 | 13.3 | **9.35** |
| DTD | Others | −27.67 | −4.92 | −10.83 | −3.44 | −11.44 | −3.01 | −7.33 | −1.76 | −12.08 | −3.71 | 1.61 | 0.01 | −1 | 0.19 | **1.71** | 0.53 |
| | DTD | 21.44 | 4.04 | 16.7 | 1.33 | 31.91 | 4.89 | 2.13 | 0.05 | 26.38 | 0.9 | 0.16 | 2.45 | **33.14** | **6.12** | 0.74 | 3.19 |
| EuroSAT | Others | −20.93 | −6.7 | −14.53 | −5.44 | −14.87 | −5.72 | −10.76 | −3.75 | −3.3 | −2.58 | −0.82 | −0.85 | 1.56 | 0.92 | **2.13** | **1.32** |
| | EuroSAT | 52.11 | 3.09 | 51.96 | 3.26 | **52.37** | 3.2 | 30.93 | **3.31** | 52.19 | 2.91 | 28.04 | 2.67 | 52.15 | 2.98 | 17.02 | 2.67 |
| GTSRB | Others | −32.71 | −8.5 | −16.38 | −3.76 | −23.89 | −5.9 | −11.65 | −0.94 | −14.29 | −3.05 | 1.62 | 1.18 | −0.33 | 1.07 | **2.26** | **1.27** |
| | GTSRB | 61.47 | **12.64** | 58.8 | 12.4 | 66.1 | 11.96 | 18.42 | 11.9 | 64.74 | 10.8 | 9.19 | 8.27 | **66.27** | 12.03 | 19.33 | 11.2 |
| MNIST | Others | −32.14 | −8.64 | −24.87 | −7.53 | −28.80 | −8.76 | −35.1 | −6.02 | −5.92 | −2.03 | 0.54 | 1.49 | −0.14 | 1.8 | −0.42 | **2.64** |
| | MNIST | 51.19 | 0.88 | 50.56 | 0.98 | **51.28** | 1.02 | 21.34 | 0.98 | 51.02 | 0.8 | 5.51 | 0.5 | 51.18 | 0.98 | 8.35 | 0.83 |
| RESISC45 | Others | −17.92 | −5.42 | −8.47 | −3.32 | −10.71 | −3.79 | −4.81 | −2.27 | −0.8 | −0.91 | 2.46 | 0.66 | 2.08 | 0.41 | **3.04** | 0.9 |
| | RESISC45 | 33.49 | **3.98** | 30.4 | 3.94 | **35.1** | 3.9 | 13.65 | 3.68 | 33.14 | 2.19 | 8.14 | 2.25 | 34.76 | 3.37 | 10.92 | 3.25 |
| SVHN | Others | −28.91 | −10.74 | −21.91 | −10.40 | −23.44 | −11.12 | −25.34 | −8.73 | −7.07 | −2.78 | −11.78 | −2.11 | 1.14 | −0.68 | −2.72 | −0.29 |
| | SVHN | 65.0 | 31.92 | 64.38 | 31.86 | **65.58** | 31.79 | 51.28 | **31.98** | 64.84 | 31.02 | 53.39 | 31.04 | 65.25 | 31.36 | 50.92 | 31.38 |
| ImageNet | Others | −5.92 | −1.6 | −5.13 | −1.39 | −5.28 | −1.26 | −4.96 | −0.87 | −1.2 | −0.24 | −0.77 | −0.1 | −0.03 | 0.13 | 0.4 | **0.35** |
| | ImageNet | 13.69 | **6.16** | 14.52 | 6.08 | **14.93** | 5.92 | 8.97 | 6.02 | 12.9 | 4.8 | 7.73 | 4.68 | 14.82 | 6.04 | 9.69 | 6.11 |

Table 1: **$\Delta_{\text{ZS}}$ and $\Delta_{\text{LP}}$ for models fine-tuned on 10 image classification datasets.** For each dataset, the first row shows $\Delta_{\text{ZS}}$ and $\Delta_{\text{LP}}$ averaged over the 8 datasets different from the fine-tuning dataset. The second row shows the same on the respective fine-tuned test sets. Best results shown in bold.

complete fine-tuning objective then becomes:

$$\mathcal{L}_{\text{LDIFS}} = \mathcal{L}_{\text{CE}} + \lambda_{\text{LDIFS}} \cdot d(\theta_{v(t)}, \theta_{v(0)}, \mathcal{D}_{\text{train}}) \quad\quad (4)$$

where $\mathcal{D}_{\text{train}}$ is the training set and $\lambda_{\text{LDIFS}}$ is the regularization coefficient. Note that similar to L2SP, we can initialize the linear head with both zero-shot weights or linear probe weights leading to two variants: ZS-init-LDIFS and LP-init-LDIFS.

**LP-init-LDIFS minimizes concept forgetting with competitive fine-tuned performance.** We evaluate LDIFS on the same setting as in §3 and present the $\Delta_{\text{LP}}\%$ over fine-tuning in Figure 5 (additional results presented in Appendix C). Furthermore, in Table 1, we report the $\Delta_{\text{ZS}}$ and $\Delta_{\text{LP}}$ of the fine-tuned models for 10 different image classification tasks. For each task, we report $\Delta_{\text{ZS}}$ and $\Delta_{\text{LP}}$ averaged over the remaining 8 tasks in the first row. For ImageNet, we leave out CIFAR-10/100 when calculating performance over other tasks since all of CIFAR-10/100's classes are within the set of ImageNet classes. In the second row, we show $\Delta_{\text{ZS}}$ and $\Delta_{\text{LP}}$ on the test set of the fine-tuning task itself. Full set of results are in Table 6 of the Appendix. Our observations are as follows.

1. From Figure 5 and Table 1, we note that both ZS-init-LDIFS and LP-init-LDIFS are competitive with other fine-tuning baselines in terms of accuracy on the downstream fine-tuning task itself.

2. ***Both the LDIFS baselines, in particular LP-init-LDIFS, significantly minimize concept forgetting over the course of fine-tuning***. This is evident from Figure 5 which show their noticeably higher $\Delta_{\text{LP}}$ on other downstream tasks over the course of fine-tuning. It is also apparent from their consistently high $\Delta_{\text{ZS}}$ and $\Delta_{\text{LP}}$ scores in Table 1. Particularly, LP-init-LDIFS gets the highest average $\Delta_{\text{LP}}$ on other tasks for 7 out of 10 fine-tuning cases. This indicates a significantly minimized level of concept forgetting. We have similar conclusions even on a CLIP RN50 model and a FLAVA Singh et al. (2022) ViT-B/16 model in Appendix C.7.

3. From Table 1 and Table 6, we also observe that in 8 out of 10 fine-tuning tasks, LP-init-LDIFS results in a positive average $\Delta_{\text{LP}}$ on other tasks, thereby generally achieving a positive forward transfer. The two exceptions are when the finetuning tasks $\mathcal{D}_{\text{ft}}$ are Stanford Cars and SVHN, in which case the average $\Delta_{\text{LP}}$ on other tasks is negative. This observation also highlights how the ordering of fine-tuning tasks can also impact forward transfer. For instance, from Table 6, when fine-tuning on EuroSAT, LP-init-LDIFS achieves a $\Delta_{\text{LP}}$ of $+3.03$ on SVHN, whereas in the reverse scenario, fine-tuning on SVHN leads to a negative $\Delta_{\text{LP}}$ of $−0.78$ on EuroSAT.

Figure 6: $\mathcal{A}_{LP}$ for sequence: SVHN → CIFAR10 → RESISC45 (left 3 plots) and ii) SVHN → CIFAR100 → RESISC45 (right 3 plots). Vertical red line indicates a switch in the fine-tuning tasks.

| Dataset | | | Fine-tuning baselines | | | | | | | | | | | | | | | | | | | | | |
| Fine-tune | Eval | | FLYP | | | FLYP-CE | | | ZS-init-CE | | | LP-init-CE (LP-FT) | | | ZS-init-L2SP | | | LP-init-L2SP | | | ZS-init-LDIFS (Ours) | | | LP-init-LDIFS (Ours) | |
| | | $\Delta_{ZS}(\uparrow)$ | $\Delta_{LP}(\uparrow)$ | $\mathcal{A}_{LP}(\uparrow)$ | $\Delta_{ZS}(\uparrow)$ | $\Delta_{LP}(\uparrow)$ | $\mathcal{A}_{LP}(\uparrow)$ | $\Delta_{ZS}(\uparrow)$ | $\Delta_{LP}(\uparrow)$ | $\mathcal{A}_{LP}(\uparrow)$ | $\Delta_{ZS}(\uparrow)$ | $\Delta_{LP}(\uparrow)$ | $\mathcal{A}_{LP}(\uparrow)$ | $\Delta_{ZS}(\uparrow)$ | $\Delta_{LP}(\uparrow)$ | $\mathcal{A}_{LP}(\uparrow)$ | $\Delta_{ZS}(\uparrow)$ | $\Delta_{LP}(\uparrow)$ | $\mathcal{A}_{LP}(\uparrow)$ | $\Delta_{ZS}(\uparrow)$ | $\Delta_{LP}(\uparrow)$ | $\mathcal{A}_{LP}(\uparrow)$ | $\Delta_{ZS}(\uparrow)$ | $\Delta_{LP}(\uparrow)$ | $\mathcal{A}_{LP}(\uparrow)$ |
|---|---|---|---|---|---|---|---|---|---|---|---|---|---|---|---|---|---|---|---|---|---|---|---|---|---|
| SVHN → C10 → R45 | SVHN | −24.63 | −7.06 | 90.3 | −19.45 | −5.77 | 91.61 | −14.8 | −7.13 | 90.29 | −19.66 | −6.46 | 90.97 | −14.37 | −5.41 | 91.01 | −11.57 | −4.53 | 91.93 | **−0.68** | **−0.43** | 96.66 | −1.36 | −0.41 | **96.68** |
| | CIFAR10 | −6.6 | −3.16 | 94.61 | −3.4 | −1.92 | 95.45 | −5.56 | −2.31 | 95.25 | −3.35 | −1.57 | 96.31 | −2.39 | −1.22 | 96.33 | **−0.23** | −0.25 | 97.26 | −0.61 | −0.26 | 97.18 | −0.55 | **−0.21** | **97.41** |
| | RESISC45 | 29.84 | **4.06** | 95.33 | 29.54 | 3.89 | 95.16 | 34.94 | 4.0 | 95.3 | −3.78 | 2.98 | 94.29 | 33.43 | 2.97 | 94.24 | 4.59 | 2.16 | 93.44 | 34.68 | 3.94 | 95.33 | 10.98 | 3.7 | 95.0 |
| | Others | −23.69 | −6.59 | 78.91 | −13.76 | −4.9 | 81.2 | −14.2 | −5.08 | 80.91 | −16.74 | −4.24 | 82.13 | −5.21 | −1.82 | 85.27 | −1.83 | −0.01 | 86.89 | −2.23 | −0.3 | 86.52 | **0.19** | **0.1** | **87.08** |
| SVHN → C100 → R45 | SVHN | −26.67 | −7.75 | 89.64 | −18.36 | −6.71 | 90.65 | −18.19 | −7.28 | 90.05 | −6.96 | −2.73 | 94.42 | −27.9 | −11.36 | 85.18 | −15.38 | −6.12 | 90.42 | −2.74 | −1.5 | 95.47 | **−2.26** | **−0.65** | **96.32** |
| | CIFAR100 | −23.0 | −8.85 | 79.22 | −16.23 | −6.38 | 81.55 | −16.69 | −7.18 | 81.08 | −11.56 | −3.04 | 82.63 | −7.63 | −3.46 | 84.2 | −1.87 | −0.88 | 85.72 | −2.84 | −0.99 | 86.45 | **−2.51** | **−0.3** | **86.54** |
| | RESISC45 | 30.68 | **4.38** | 95.68 | 30.95 | 3.9 | 95.21 | **35.0** | 4.13 | 95.4 | −2.75 | 2.51 | 93.81 | 33.02 | 2.79 | 94.13 | 2.84 | 1.9 | 93.21 | 34.89 | 4.29 | 95.59 | 9.19 | 3.83 | 95.11 |
| | Others | −18.63 | −5.23 | 83.09 | −9.04 | −4.68 | 83.97 | −10.7 | −4.65 | 83.76 | −11.32 | −4.02 | 85.14 | −3.19 | −1.61 | 87.52 | −2.06 | −0.37 | 89.04 | −1.92 | −0.56 | 88.36 | **−0.6** | **−0.23** | **89.12** |
| SVHN → Cars → R45 | SVHN | −3.62 | −1.26 | 96.07 | −4.79 | −1.51 | 95.76 | −2.89 | −1.45 | 95.93 | −0.94 | −0.76 | 96.58 | −2.2 | −1.14 | 95.42 | −0.51 | −0.44 | 95.98 | **−0.49** | **−0.49** | 96.56 | −2.11 | −0.17 | **96.9** |
| | Cars | −8.79 | −3.61 | 81.21 | −4.3 | 76.87 | −9.91 | −4.18 | 76.96 | −22.62 | −8.36 | 71.6 | −4.14 | −0.88 | 81.18 | −7.06 | −0.4 | 81.82 | **−1.12** | −0.61 | 82.89 | −2.71 | **0.47** | **84.23** |
| | RESISC45 | 31.89 | **4.13** | 95.43 | 29.37 | 3.81 | 95.08 | 34.73 | 3.89 | 95.17 | −1.16 | 3.0 | 94.35 | 33.22 | 2.83 | 94.13 | 4.03 | 2.13 | 93.43 | **35.0** | 3.94 | 95.22 | 9.13 | 3.73 | 95.27 |
| | Others | −20.44 | −6.68 | 81.91 | −14.66 | −5.48 | 83.2 | −12.64 | −4.93 | 83.38 | −19.04 | −4.51 | 84.39 | −7.06 | −2.91 | 85.74 | −8.13 | −1.67 | 87.15 | −1.25 | −0.1 | 88.75 | **0.9** | **0.23** | **89.39** |

Table 2: $\Delta_{ZS}$, $\Delta_{LP}$ and $\mathcal{A}_{LP}$ for models fine-tuned on (SVHN, C10, RESISC45), (SVHN, C100, RESISC45) & (SVHN, Cars, RESISC45) sequences. The first 3 rows show performance on fine-tuned tasks and the third row shows averaged performance on the remaining 6 datasets.

| Dataset | | | Fine-tuning baselines | | | | | | | | | |
| Fine-tune | Eval | | LwF | | LFL | | iCaRL | | D+R | | ZSCL | | LP-init-LDIFS (Ours) |
| | | $\Delta_{LP}(\uparrow)$ | $\mathcal{A}_{LP}(\uparrow)$ | $\Delta_{LP}(\uparrow)$ | $\mathcal{A}_{LP}(\uparrow)$ | $\Delta_{LP}(\uparrow)$ | $\mathcal{A}_{LP}(\uparrow)$ | $\Delta_{LP}(\uparrow)$ | $\mathcal{A}_{LP}(\uparrow)$ | $\Delta_{LP}(\uparrow)$ | $\mathcal{A}_{LP}(\uparrow)$ | $\Delta_{LP}(\uparrow)$ | $\mathcal{A}_{LP}(\uparrow)$ |
|---|---|---|---|---|---|---|---|---|---|---|---|---|---|
| SVHN → C10 → R45 | SVHN | −3.81 | 90.48 | −3.21 | 91.9 | −3.67 | 91.62 | −2.78 | 93.3 | −3.23 | 92.7 | **−0.41** | **96.68** |
| | CIFAR10 | −2.9 | 93.9 | −2.32 | 94.88 | −2.1 | 95.17 | −1.9 | 95.41 | −1.6 | 95.82 | **−0.21** | **97.41** |
| | RESISC45 | 3.1 | 94.22 | 2.98 | 93.9 | 2.83 | 93.72 | 3.68 | 94.94 | 3.62 | 94.89 | **3.7** | **95.0** |
| | Others | −4.2 | 80.73 | −3.76 | 81.31 | −4.11 | 80.78 | −3.2 | 81.86 | −2.8 | 83.1 | **0.1** | **87.08** |
| SVHN → C100 → R45 | SVHN | −4.34 | 89.48 | −4.08 | 90.29 | −4.31 | 90.97 | −3.23 | 92.3 | −3.92 | 91.81 | **−0.65** | **96.32** |
| | CIFAR100 | −3.25 | 83.24 | −3.01 | 83.95 | −3.13 | 84.06 | −2.6 | 84.82 | −2.13 | 85.07 | **−0.3** | **86.54** |
| | RESISC45 | 3.21 | 93.8 | 3.62 | 94.91 | 3.54 | 94.87 | 3.71 | 95.08 | 3.65 | 94.96 | **3.83** | **95.11** |
| | Others | −4.11 | 81.73 | −3.8 | 82.04 | −4.02 | 81.62 | −3.43 | 82.17 | −3.11 | 82.86 | **−0.23** | **89.12** |
| SVHN → Cars → R45 | SVHN | −3.64 | 91.43 | −2.92 | 92.74 | −3.13 | 91.75 | −2.84 | 92.86 | −2.72 | 92.98 | **−0.17** | **96.9** |
| | Cars | −2.79 | 81.69 | −2.64 | 81.82 | −2.8 | 81.7 | −2.12 | 82.11 | −1.84 | 82.68 | **0.47** | **84.23** |
| | RESISC45 | 3.34 | 93.92 | 3.55 | 94.96 | 3.58 | 94.97 | 3.72 | 95.19 | 3.63 | 95.04 | **3.73** | **95.27** |
| | Others | −4.07 | 81.63 | −3.6 | 82.24 | −3.89 | 81.88 | −3.12 | 82.73 | −2.8 | 83.1 | **0.23** | **89.39** |

Table 3: $\Delta_{LP}$ and $\mathcal{A}_{LP}$ comparing LDIFS with 5 distillation based continual learning methods.

## 5 A NUDGE TOWARDS CONTINUAL FINE-TUNING

Our results above indicate that fine-tuning on LP-init-LDIFS can make the foundation model learn new downstream information without forgetting pre-trained concepts. A natural question that then arises is: *Can we fine-tune on a sequence of downstream tasks without forgetting concepts?* For an ideal fine-tuning method, the final model should attain state-of-the-art performance on all fine-tuned tasks while still maintaining its pre-trained knowledge. We try to empirically answer this question by training on 3 sequences of 3 tasks each: **a)** SVHN → C10 → R45, **b)** SVHN → C100 → R45 and **c)** SVHN → Cars → R45. This setup is similar to continual learning Chaudhry et al. (2018); Rebuffi et al. (2017); Lopez-Paz & Ranzato (2017) but for foundation models. Due to their impressive ZS and LP performance on a lot of general downstream tasks, continual fine-tuning of foundation models is relatively unexplored in the literature. Nonetheless, as stated in §1, this is an important problem to investigate from the perspective of updating a foundation model with new previously unknown knowledge without making it forget previously trained concepts.

In order to quantify concept forgetting on a task $\mathcal{D}$ over a sequence of models $f_{\theta_0}, ..., f_{\theta_k}$ fine-tuned on the task sequence $\mathcal{D}_1 \to \mathcal{D}_2, \to ..., \mathcal{D}_k$, we follow intuitions from Chaudhry et al. (2018), and extend the $\Delta_{LP}$ from Equation (2) as follows:

$$\Delta_{LP}(\mathcal{D}, f_{\theta_k}, \{f_{\theta_0}, f_{\theta_1}, ..., f_{\theta_{k-1}}\}) = \mathcal{A}_{LP}(f_{\theta_k}, \mathcal{D}) - \max_{i \in \{0,...,k-1\}} \mathcal{A}_{LP}(f_{\theta_i}, \mathcal{D}). \quad (5)$$

$\Delta_{ZS}$ can also be defined in a similar manner. Note that we find the difference in ZS or LP performance between the final fine-tuned model $f_{\theta_k}$ and the model having the maximum ZS or LP performance in the sequence $\{f_{\theta_0}, f_{\theta_1}, ..., f_{\theta_{k-1}}\}$. It is trivial to see that on a single task, eq. (5) reduces to eq. (2). In Table 2, we present the $\Delta_{ZS}$, $\Delta_{LP}$ and $\mathcal{A}_{LP}$ for the sequence setup for all fine-tuning baselines. For each task sequence, we report the performance on the datasets in the sequence itself in the first two rows and in the third row, we report the average performance on all other datasets.

From Table 2, we observe: **a)** *for all 3 sequences, LP-init-LDIFS has minimal concept forgetting on tasks which appear earlier in the fine-tuning sequence, as well as other datasets which are not used for fine-tuning*, **b)** at the same time, *LP-init-LDIFS is very competitive on accuracy outperforming L2SP on the last fine-tuning task*, i.e., R45. This observation is further complemented in Figure 6 where we can see LP-init-LDIFS to have minimal forgetting on prior tasks like SVHN and C10/100 without compromising on performance on the last fine-tuned task, RESISC45.

## 6 RELATED WORK

**Fine-tuning in foundation models**: There are many flavours of fine-tuning which improve on CLIP's zero-shot performance. A popular approach is prompt tuning where, methods like CoOp Zhou et al. (2022b), CoCoOp Zhou et al. (2022a), TPT Shu et al. (2022), Chain of Thought prompt tuning Ge et al. (2023), instead of hand-crafting prompts, learn the prompts specific to the fine-tuning task. Another category of methods uses adapters: CLIP-Adapter Gao et al. (2021), Tip-Adapter Zhang et al. (2021), Prompt Adapter Sun et al. (2023), SVL-Adapter Pantazis et al. (2022), which works on the principle of combining pre-trained features with a small non-linear adapter network where the adapter is fine-tuned on the downstream task at hand. Finally, along with end-to-end fine-tuning, there are weight space interpolation methods like Wise-FT Wortsman et al. (2022b), PAINT Ilharco et al. (2022b), task arithmetic Ilharco et al. (2022a) and model soups Wortsman et al. (2022a) which look at interpolating between pre-trained and fine-tuned models in the weight space in order to achieve the best of both worlds in terms of downstream task performance and robustness to distribution shift.

In Table 5 of the appendix, we compare linear probing, CoOp, CLIP-Adapter, Tip-Adapter and the classic end-to-end fine-tuning method, ZS-init-CE on downstream task performance on all 9 image classification tasks. Not only do we find end-to-end fine-tuning to consistently outperform adapters and prompt tuning, but when there is a big performance gap between the linear probe and end-to-end fine-tuning, adapters and prompt tuning don't bridge that gap well. This observation reaffirms the importance of end-to-end fine-tuning and thereby necessitates the investigation of better end-to-end fine-tuning methods for tasks where the pre-trained model's performance is sub-par.

**Knowledge Distillation & Continual Learning**: The LDIFS regularizer can be studied from the lens of knowledge distillation, an approach which has been applied to different ends like calibration He et al. (2023), pre-training Lee et al. (2022), transfer learning Zhou et al. (2022c), robust fine-tuning Mao et al. (2022) and continual learning Li & Hoiem (2017); Jung et al. (2016); Rebuffi et al. (2017); Hou et al. (2018). The main difference between these works and ours is that unlike classic distillation methods which use either the last layer features Jung et al. (2016) or the logits Li & Hoiem (2017); Rebuffi et al. (2017) directly for distillation, we concatenate features from shallower layers in the network when computing the LDIFS regularizer. This proposed modification to distillation turns out to be crucial for performance.

To empirically study this, we first compare performance of LP-init-LDIFS with 5 well-known distillation based continual learning baselines: LwF Li & Hoiem (2017), LFL Jung et al. (2016), iCARL Rebuffi et al. (2017), Distillation + Retrospection (D+R) Hou et al. (2018) and ZSCL Zheng et al. (2023) on our sequence setup. Results are in Table 3. Clearly, LP-init-LDIFS performs better than all other baselines, both in preventing forgetting as well as obtaining best performance on the last task. This is evident from its consistently high $\Delta_{ZS}$ and $\Delta_{LP}$ on all 3 sequences. To further investigate the importance of distilling from earlier features, in Figure 11 of the appendix, we compare performance of LDIFS when distilling from just last layer features (we call this ablation LL). Thes plots provide further evidence that using just the last layer features is not nearly as performant as using the earlier features in the feature vector. This indicates that learned concepts can be encoded in shallower layers of the network which makes distilling from them crucial. Finding which layers contain more pre-trained information for a downstream task is an interesting area of further research.

## 7 CONCLUSION

We explore how end-to-end fine-tuning approaches often applied on foundation models can, in turn, significantly damage the model's ability to recognize concepts outside the fine-tuning task at hand, a phenomenon which we call concept forgetting. Such an effect is undesirable particularly for foundation models which have been specifically trained on millions of samples to encode information on a vast number of real-world concepts. However, we find that among fine-tuning methods, L2SP, which keeps the model near the pre-trained model in its parameter space suffers less from concept forgetting. From insights gained by analyzing L2SP, we also find that feature space distance provides a better definition for the vicinity of the pre-trained model as the pre-trained concepts are indeed encoded in the feature space. Our proposed new regularizer, LDIFS, encourages the features of fine-tuned model to be in the vicinity of the pre-trained ones. Through extensive evaluation on 10 different downstream classification tasks, as well as a continual fine-tuning setup of 3 different sequences of 3 tasks each, we showed that LDIFS significantly alleviates concept forgetting without impeding downstream task performance.

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
