# OpenReview forum: "Fine-tuning can cripple foundation models; preserving features may be the solution"
_ICLR.cc/2024/Conference — ICLR 2024 Conference Withdrawn Submission_

### Official Review · Reviewer_Deyx · 2023-10-28

**Soundness:** 3 good
**Presentation:** 3 good
**Contribution:** 2 fair
**Rating:** 3
**Confidence:** 4

**Summary:**

The authors propose an extensive study to test the ability of several fine-tuning algorithms to preserve the initial knowledge of the fine-tuned models. Based on their results, they propose to add a distillation-based regularization term, aiming to prevent the so-called catastrophic forgetting. The experimental results over their approach shows the algorithm is capable of preserving the model's knowledge, with almost no accuracy drop in the fine-tuned dataset.

**Strengths:**

**originality**: this is the first time a have seem such a exhaustive analysis regarding the forgetting problem of fine-tuned models.

**quality**:  the authors provide an exhaustive and comprehensive analysis over multiple fine-tuning models, showing both their advantages and limitations.

**clarity**: the paper is easy to read and to follow. Furthermore, their regularization term is very easy to implement.

**significance**: the concept of catastrophic forgetting is of special interest for modern machine learning models that requires online training.

**Weaknesses:**

**originality**: as the authors already mentioned, their idea is extremely similar to some distillation techniques, specially with the concept of self-distillation [1]. Besides that, there were several studies focused on the idea of preserving the model's prior knowledge. This field is often referred as Lifelong Learning [2] (a subsection of the incremental learning theory). [2] also provides a knowledge distillation approach t o solve this problem, although I must say it is not the same to the one provided by the authors. I suggest the authors to introduce in their paper some of these approaches, stating the advantages of their solution.

**significance**: due to the originality issue, it is not clear that this paper provides enough innovations to be accepted in a venue of this kind.


[1] Zhang, L., Bao, C., & Ma, K. (2021). Self-distillation: Towards efficient and compact neural networks. IEEE Transactions on Pattern Analysis and Machine Intelligence, 44(8), 4388-4403.

[2] Hou, S., Pan, X., Loy, C. C., Wang, Z., & Lin, D. (2018). Lifelong learning via progressive distillation and retrospection. In Proceedings of the European Conference on Computer Vision (ECCV) (pp. 437-452).

**Questions:**

- What are the advantages of your algorithm compared with other lifelong learning approaches (like the one mentioned in the weaknesses section)?

**Details Of Ethics Concerns:**

It would be of interest to know if the author's proposal introduces (or mitigates) possible bias that are included in the prior knowledge.

---

> ### Author Response · Authors · 2023-11-18
> **Advantages of LDIFS over Continual/Lifelong Learning Approaches**
>
> Thank you for taking the time to review our paper and for your feedback! We provide our response below.
>
> We would like to highlight that we design LDIFS to be an end-to-end fine-tuning method which also minimizes concept forgetting. Like all end-to-end fine-tuning methods, obtaining state-of-the-art accuracy on the downstream task at hand is a priority for LDIFS. This is a well-known problem with classic continual learning methods.
>
> **LDIFS is different from classic distillation.**
> * Most knowledge-distillation methods used in continual learning (CL)  [A1, A2, A3, A4, A5] distill either from the logits of an expert model [A1, A3, A4] or from its last layer features [A2, A5].
> * LDIFS distills not only from the last layer features but also earlier shallower features along the network architecture (see Appendix B for details). We find this modification to distillation to be crucial to LDIFS’s performance.
> * To empirically see this, we compare with an ablation on LDIFS where we distill only from the last layer features. We show these results in Fig 11 of Appendix C.3 and find that using just last layer features is not nearly as performant as distilling from earlier features as well.
>
> **LDIFS outperforms distillation based CL and E2E fine-tuning on continual setups.**
> * We have improved our continual learning setup to 3 sequences of 3 tasks each and  compare LDIFS with 5 different distillation based CL baselines. Results are in Table 3 of the revised draft. We find LP-init-LDIFS to consistently outperform all 5 CL baselines both on preventing forgetting and obtaining the best performance on the last task in the sequence, i.e., RESISC45.
> * In addition, LDIFS minimizes concept forgetting compared to other E2E  fine-tuning baselines as well, at the same time providing competitive accuracies on the last fine-tuned task. See Table 2 in the revised draft for results.
>
> **LDIFS’s simplicity makes it easily applicable to different architectures and foundation models.**
> * The LDIFS regularizer computes an L2 distance between the features of the pre-trained and current model during fine-tuning. Hence, it does not require any architectural changes to the network, which is what Bao et al. [1] does by training shallow classifiers with attention layers on intermediate features.
> * This makes LDIFS easily scalable and applicable to larger architectures, as we show in Appendix C.7 (and Tables 8 and 9) where we study concept forgetting in CLIP RN50 and FLAVA ViT-B/16 and find LDIFS to be the most performant on these models as well.
>
> Our results indicate that a network can encode learned concepts in activations of shallower layers as well, thereby making distillation from such layers necessary to minimize concept forgetting. We have explained this in Section 6 of the revised draft and updated our results. We hope that this explanation brings to light the novelty and benefits of the LDIFS regularizer compared to other distillation based CL/lifelong learning methods.
>
> [A1] Li, Z. and Hoiem, D., 2017. Learning without forgetting. IEEE transactions on pattern analysis and machine intelligence, 40(12), pp.2935-2947.
>
> [A2] Jung, H., Ju, J., Jung, M. and Kim, J., 2018, April. Less-forgetful learning for domain expansion in deep neural networks. In AAAI 2018.
>
> [A3] Rebuffi, S.A., Kolesnikov, A., Sperl, G. and Lampert, C.H., 2017. icarl: Incremental classifier and representation learning. In CVPR 2017
>
> [A4] Hou, S., Pan, X., Loy, C.C., Wang, Z. and Lin, D., 2018. Lifelong learning via progressive distillation and retrospection. In ECCV 2018.
>
> [A5] Zheng, Z., Ma, M., Wang, K., Qin, Z., Yue, X. and You, Y., 2023. Preventing Zero-Shot Transfer Degradation in Continual Learning of Vision-Language Models. ICCV 2023.

---

> > ### Author Response · Authors · 2023-11-20
> > **A Gentle Follow-up**
> >
> > Dear reviewer Deyx,
> >
> > Thank you again for the time you took to review our work!
> >
> > We believe that we have addressed your concern on the difference and advantages of LDIFS compared to classic distillation based continual/lifelong learning methods.
> >
> > We would really appreciate it if you could have a look at our replies and let us know if you had further questions/comments.
> >
> > We hope to be able to effectively utilize the discussion phase in case you had any further questions or comments about our work.
> >
> > Thank you!

---

> > > ### Author Response · Authors · 2023-11-21
> > > **Utilizing the discussion phase**
> > >
> > > Dear reviewer Deyx,
> > >
> > > Thank you for your review on our work!
> > >
> > > We would appreciate it if you could please have a look at our replies and we are more than happy to get back to you if you have further questions or comments.
> > >
> > > We believe that this work holds value for the research community, as you and the other reviewers have also pointed out, and we want to make full use of the discussion phase to engage with you and clarify any further concerns that you may have.
> > >
> > > The author-reviewer discussion period ends tomorrow and so we will really appreciate if you could kindly have a look at our reply before that.
> > >
> > > Thank you very much again for your time and efforts on reviewing our paper!

---

> > > > ### Comment · Reviewer_Deyx · 2023-11-22
> > > > **Response to authors**
> > > >
> > > > Dear authors,
> > > >
> > > > I would like to thank the authors for some clarifications regarding the paper. Unfortunately, I still think the novelty of this contribution is minimal for a venue of this kind.
> > > >
> > > > Besides that, I will urge the authors to increase the size of both tables and images, as they are extremely hard to see in its current form.

---

### Official Review · Reviewer_2XG5 · 2023-10-29

**Soundness:** 3 good
**Presentation:** 3 good
**Contribution:** 3 good
**Rating:** 8
**Confidence:** 4

**Summary:**

The paper analyzes the concept of forgetting phenomenon when fine-tuning pre-trained models.
To solve such a forgetting problem, they propose a method called LDIFS which adds l_2 distance to the feature space.
The method can significantly reduce the concept forging on several downstream datasets.

**Strengths:**

1. The writing and presentation is fluent.
2. The analysis of forgetting on various datasets is comprehensive.
3. The method is easy to understand and reasonable.

**Weaknesses:**

1. The baselines to be compared are limited. The experiments mainly compare with baselines with additional regularization to prevent forgetting. But works like Wise-FT mentioned in the related work also show good and even better performance on preventing forgetting which is not compared in this paper.

2. It is not direct to tell how good the proposed method is from Table 1.
It will be more clear if the results in Table 1 can be visualized in some other direct ways, e.g., take the car's value as the x-axis and others as the y-axis, then plot the performance of all methods on a picture.

**Questions:**

1. I think it will be better to add some experiments that compare the methods like WISE-FT or analyze whether the proposed method can combined with the WISE-FT to achieve a better performance.
2. As we know, the hyperparameters of the regularization weights \lambda may affect the final performance of the fine-tuned model. But I do not find any discussion about that. It will be better to add such kind of analysis about the hyperparameters.

---

> ### Author Response · Authors · 2023-11-18
> **WiseFT & Visualizations**
>
> Thank you for taking the time to review our paper and for your feedback! We provide our response below.
>
> **“The experiments mainly compare with baselines with additional regularization to prevent forgetting.”**
>
> * To motivate the importance of end-to-end fine-tuning in foundation models, we compared with both *prompt tuning* and *adapter* baselines in Appendix C.4 and Table 5.
> * In the revised draft, we have also provided comparisons with *5 continual learning baselines* in Table 3 of the main paper.
> * We have improved our continual learning setup from 2 sequences of 2 tasks to 3 sequences of 3 tasks each (see Section 5 of the revised draft).
> * We have also incorporated Wise-FT as an ablation as we describe next.
>
>
> **“I think it will be better to add some experiments that compare the methods like WISE-FT or analyze whether the proposed method can combined with the WISE-FT”**
>
> Thank you for this suggestion.
> * Wise-FT is a weight interpolation method which requires a pre-trained and fine-tuned model to work. Thus, it can be applied on every end-to-end fine-tuning baseline in our paper.
> * In Appendix C.5 and Table 4 of the revised draft, we show the effect of applying Wise-FT on minimizing concept forgetting when fine-tuned on 3 tasks: CIFAR-10, EuroSAT and SVHN.
> * We find that applying Wise-FT consistently reduces concept forgetting across all the fine-tuning baselines. However, the order of performance even after applying Wise-FT remains broadly consistent with LDIFS outperforming other baselines.
>
>
> **"It will be more clear if the results in Table 1 can be visualized in some other direct ways, e.g., take the car's value as the x-axis and others as the y-axis, then plot the performance of all methods on a picture."**
>
> * In Appendix C.6 and Figure 12 of the revised draft, we provide the visualization you ask for. For each downstream task and each baseline, we plot its test accuracy on the x-axis and the average test accuracy on all other tasks on the y-axis.
> * We find LDIFS baselines to consistently lie around the top right corner of these plots indicating high accuracies both on the downstream task itself while preserving performance on other tasks.
>
>
> **“As we know, the hyperparameters of the regularization weights \lambda may affect the final performance of the fine-tuned model. I do not find any discussion about that. It will be better to add such kind of analysis about the hyperparameters.”**
>
> * In the previous draft, we did discuss the choice of $\lambda$ in the second paragraph of Appendix C.3, “Choosing $\lambda_{\mathrm{LDIFS}}$”.
> * We perform a grid search over {0.01, 0.05, 0.1, 0.5, 1, 10, 100} and use the lambda which produces the highest accuracy on the validation set of the downstream task to be fine-tuned on. In our experiments, we consistently find the best $\lambda$ value to be 10 for all datasets, so all the LDIFS results we report are with $\lambda = 10$.
> * In the revised draft, we have moved this paragraph to Appendix A (Training Details).

---

> > ### Author Response · Authors · 2023-11-20
> > **A Gentle Follow-up**
> >
> > Dear reviewer 2XG5,
> >
> > Thank you again for the time you took to review our work!
> >
> > We believe that we have clarified your concerns regarding the baselines we compare with, adding Wise-FT as an ablation, your suggested alternative visualization for the Table 1 results and choosing the $\lambda$ hyperparameter.
> >
> > We would really appreciate it if you could have a look at our replies and let us know if you had further questions/comments.
> >
> > We hope to be able to effectively utilize the discussion phase in case you had any further questions or comments about our work.
> >
> > Thank you!

---

> > ### Comment · Reviewer_2XG5 · 2023-11-21
> > **Reply**
> >
> > Thank the author for the detailed rebuttal.
> >
> > I think many experiments' concerns have been addressed and I will change the score from 6 to 7. (Since there is no 7, I just choose the 8).

---

> > > ### Author Response · Authors · 2023-11-21
> > > **Thank you!**
> > >
> > > Thank you very much for acknowledging our response and raising your score! We are very glad that we were able to clarify your concerns.

---

### Official Review · Reviewer_VWxz · 2023-11-01

**Soundness:** 3 good
**Presentation:** 3 good
**Contribution:** 2 fair
**Rating:** 5
**Confidence:** 3

**Summary:**

The paper proposes a regularization method to reduce the "concept forgetting" when finetuning a pre-trained foundational model with limited data. It first shows that finetuning CLIP on a downstream task degrades the model's recognition of concepts outside the target downstream task. To mitigate this issue, it proposes a simple regularization loss, which encourages to preserve the original CLIP during finetuning. Specifically, it empirically shows that minimizing the l-2 distance to the original model in the feature space (LDIFS) is consistently better than minimizing the l-2 distance in the parameter space, resulting in less "concept forgetting". Experiments are done for 6 different finetuning methods on 9 downstream classification tasks.

**Strengths:**

- This paper studies an interesting problem, i.e., "concept forgetting" of a pre-trained foundational model when it is finetuned to relatively small data of a target downstream task.
- The idea to minimize the distance to the original model in the feature space is simple but works well. In the experiments, it consistently shows better performance than the alternative to minimize the distance in the parameter space.
- In particular, the paper provides extensive experiments on all combinations of 6 different finetuning methods and 9 classification datasets.
- The paper is well written. It was easy to follow.

**Weaknesses:**

1. Although the paper reported results of many experiments, I think the impact could be somewhat limited because all downstream tasks are classification with relatively small-sized training data. I am not sure if the observation in this paper (that l2-distance in features space is better metric for regularization) will generalize to other practical settings when there are medium-sized training data that covers many concepts. For example, there are observations that CLIP finetuned on existing detection dataset achieves good performance on open vocabulary detection [1], which implies that the finetuned CLIP can recognize concepts unseen in the finetuning dataset instead of forgetting them, possibly by interpolating the seen concepts. Also, when downstream task is not a classification task but is less similar to pre-training contrastive loss, such as detection, the stability-plasticity trade-off could be worse and the effect of regularization methods could be different. I think it would be interesting to do similar experiments in the paper to such other downstream tasks.

2. I think "catastrophic forgetting" and "concept forgetting" address the same problem, as also mentioned in the paper. Since catastrophic forgetting have been studied in many continual learning approaches, it would be nice to discuss them in more detail in the related work chapter and also add comparison with some representative methods [2] in the experiments as strong baselines when they are applicable. Especially, the idea to distill knowledge from the original model to the finetune model as a regularization looks similar, which has been studied for continual learning.

[1] Simple Open-Vocabulary Object Detection with Vision Transformers, ECCV 2022
[2] A continual learning survey: Defying forgetting in classification tasks, TPAMI 2020

**Questions:**

- Additional discussion and comparison with continual learning methods would be necessary
- Additional evaluation on other downstream tasks beyond classification would be nice
- Tables and figures are very small. Some of them could be merged (fig 2 and 5) or moved to the appendix.
- Finetuning with reduced learning rate for the backbone [1] could be another baseline

---

> ### Author Response · Authors · 2023-11-18
> **Continual Learning Comparisons and Introducing a Medium-sized Dataset**
>
> Thank you for taking the time to review our paper and for your feedback! We provide our response below.
>
> **“Additional discussion and comparison with continual learning methods would be necessary”**
>
> Thank you for this suggestion.
> * In the revised draft, in Section 6 (Related Work), we provide a more detailed discussion of knowledge distillation in continual learning (CL), specifically highlighting how LDIFS differs from other distillation based CL methods.
> * In Section 5, we have improved our continual learning setup from 2 sequences with 2 tasks each to 3 sequences with 3 tasks each, with Table 2 and Figure 6 reflecting these changes.
> * In Table 3 of the revised main paper, we provide a comparison with 5 different distillation based continual learning methods on our improved 3-task sequence setup. We find LDIFS to consistently outperform all competitive continual learning baselines on all task sequences.
>
> **“Although the paper reported results of many experiments, I think the impact could be somewhat limited ... I think it would be interesting to do similar experiments in the paper to such other downstream tasks.”**
>
> Thank you for mentioning this interesting point. We will answer this in two parts.
>
> *Evaluating on ImageNet, a medium sized dataset with 1000 classes*:
> * First, to study concept forgetting and the effects of LDIFS on a medium sized dataset with a larger number of concepts, we include ImageNet (with 1000 classes) as the 10th downstream classification dataset in Table 1 of the revised draft.
> * Since CIFAR-10/100 have classes within the set of ImageNet classes, we leave out CIFAR-10/100 when evaluating concept forgetting for models fine-tuned on ImageNet. As evident from Table 1, even with a 1000 class training dataset, our observations indicate both the existence of concept forgetting as well as the superiority of LP-init-LDIFS on minimizing concept forgetting.
>
> *Concept forgetting in CLIP beyond classification*
> * These is an interesting question, however, it is not obvious how we can quantify concept forgetting for CLIP when moving beyond classification to a task like object detection.
> * The reason being that there is no straightforward way to quantify performance of a pre-trained CLIP model on object detection. If there were, zero-shot or linear probe object detection would be an immediate baseline in Owl-ViT [1], a work which specifically focuses on extending CLIP like models to open-vocabulary object detection.
> * As far as recognizing concepts not seen in the fine-tuning set [1] is concerned, this observation is not very surprising. In our experiments, we observe a performance drop in classifying pre-trained concepts during fine-tuning and use this performance drop to quantify forgetting. In fact, the numbers in Table 1 indicate that the fine-tuned models are still able to recognize and classify pre-trained concepts, just not as well as the pre-trained models, thereby showing a certain level of forgetting.
>
> We do think that this point is interesting and warrants more thought to design ways to properly quantify concept forgetting when moving beyond image classification. We look forward to investigating this as future work.
>
> **"Tables and figures are very small. Some of them could be merged (fig 2 and 5) or moved to the appendix."**
>
> Thank you for this point. We intentionally kept Figures 2 and 5 separate to maintain the flow of thought and not “spoil” LDIFS before we motivate LDIFS in our paper. We will work on increasing the figure size and readability in the final draft.
>
> **"Finetuning with reduced learning rate for the backbone [1] could be another baseline."**
>
> We tried the ablation of training with an initial learning rate of 1e-7 (100x lower than 1e-5) on the encoder for fine-tuning on CIFAR-10 (C10) and SVHN using all the fine-tuning baselines. We present the results in the table below where for each baseline, baseline+lr represents the ablation with lower learning rate on the encoder. The scores with and without this ablation are mostly competitive and we didn’t notice any particular trend in the results. Hence, we chose to leave it out of the revised draft.
>
> | Dataset  | FLYP | FLYP+lr | FLYP-CE | FLYP-CE+lr | ZS-CE | ZS-CE+lr | LP-CE | LP-CE+lr | ZS-L2SP | ZS-L2SP+lr | LP-L2SP | LP-L2SP+lr | ZS-LDIFS | ZS-LDIFS+lr | LP-LDIFS | LP-LDIFS+lr |
> | ---  | --- | --- | --- | --- | --- | --- | --- | --- | --- | --- | --- | --- | --- | --- | --- | --- |
> | C10 | -3.35 | -3.12 | -1.17 | -1.21 | -1.6 | -1.54 | -0.81 | -1.03 | 1.04 | 0.96 | 1.16 | 1.09 | 1.03 | 0.97 | 1.18 | 1.20 |
> |SVHN | -10.74 | -10.28 | -10.4 | -10.36 | -11.12 | -10.97 | -8.73 | -8.29 | -2.78 | -3.01 | -2.11 | -2.15 | -0.68 | -0.74 | -0.29 | -0.22 |

---

> > ### Author Response · Authors · 2023-11-20
> > **A Gentle Follow-up**
> >
> > Dear reviewer VWxz,
> >
> > Thank you again for the time you took to review our work!
> >
> > We believe that we have addressed your concerns regarding comparisons with continual learning baselines, incorporating a medium sized dataset with more concepts and the ablation on fine-tuning with a smaller learning rate for the backbone.
> >
> > We would really appreciate it if you could have a look at our replies and let us know if you had further questions/comments.
> >
> > We hope to be able to effectively utilize the discussion phase in case you had any further questions or comments about our work.
> >
> > Thank you!

---

> > > ### Author Response · Authors · 2023-11-21
> > > **Utilizing the discussion phase**
> > >
> > > Dear reviewer VWxz,
> > >
> > > Thank you for your review on our work!
> > >
> > > We would appreciate it if you could please have a look at our replies and we are more than happy to get back to you if you have further questions or comments.
> > >
> > > We believe that this work holds value for the research community, as you and the other reviewers have also pointed out, and we want to make full use of the discussion phase to engage with you and clarify any further concerns that you may have.
> > >
> > > The author-reviewer discussion period ends tomorrow and so we will really appreciate if you could kindly have a look at our reply before that.
> > >
> > > Thank you very much again for your time and efforts on reviewing our paper!

---

### Official Review · Reviewer_cA9V · 2023-11-07

**Soundness:** 3 good
**Presentation:** 4 excellent
**Contribution:** 2 fair
**Rating:** 3
**Confidence:** 4

**Summary:**

The paper discussed a phenomenon called “concept forgetting,” where fine-tuning a large-scale pre-trained vision language model (CLIP) on a downstream task will significantly reduce the performance on tasks distinct from the particular task. The authors showed extensive empirical evidence for concept forgetting, described a solution that distills features from the pre-trained model during fine-tuning, and presented promising results on continual learning.

**Strengths:**

* The paper addressed an important and trendy topic on adapting foundation models.

* The paper is well organized. Key concepts are clearly introduced and technical details are easy to follow.

* Results on continual learning are encouraging.

**Weaknesses:**

While the phenomenon of “concept forgetting” is interesting, as the authors admitted, it is closely related to catastrophic forgetting that was described in many prior works. Yes, I agree there are some subtle differences as described in Sec 3.1, yet it is not surprising to expect “foundation models” to have a similar behavior. Indeed, this forgetting phenomenon of CLIP has been discussed in some recent papers [a, b].

The proposed solution follows the knowledge distillation framework, which has been previously considered by several prior methods to alleviate catastrophic forgetting in the context of continual learning (e.g., iCaRL and [c]). The proposed solution is conceptually similar to those prior methods. The innovation seems quite incremental.

[a] Lee et al., Do Pre-trained Models Benefit Equally in Continual Learning? WACV 2023

[b] Ding et al., Don't Stop Learning: Towards Continual Learning for the CLIP Model, aXiv 2022

[c] Li and Hoiem, Learning without Forgetting, TPAMI 2017

**Questions:**

It will be great if the authors can highlight the conceptual innovation of the paper w.r.t. those missing prior works (e.g., [a-c]).

---

> ### Author Response · Authors · 2023-11-18
> **Conceptual Innovation & Difference from Classic Knowledge Distillation**
>
> Thank you for taking the time to review our paper and for your feedback! We provide our response below.
>
> **"While the phenomenon of “concept forgetting” is interesting, ... in some recent papers [a, b]."**
>
> **"It will be great if the authors can highlight the conceptual innovation of the paper w.r.t. those missing prior works (e.g., [a-c])."**
>
> We explain the conceptual innovation of our work as follows.
>
> *Focus on E2E fine-tuning*
> * End-to-end (E2E) fine-tuning of foundation models is an active area of research and most works on this topic primarily focus on maximizing downstream task performance.
> * Our paper is a take on i) how the latest E2E fine-tuning methods (like FLYP) applied to foundation models can lead to the forgetting of pre-trained concepts and ii) why minimizing concept forgetting should be an important additional criterion when designing fine-tuning methods, especially for foundation models which are a knowledge store of many pre-trained concepts.
> * Unlike [a, b], we don’t start our investigation with well-known continual learning baselines, but with well-known E2E fine-tuning methods. Primary reason for this being that we do not want to devise an approach that compromises with the performance on the downstream task, a common observation among existing continual learning approaches.
>
> *A New E2E Fine-tuning method*
> * Catastrophic forgetting has been studied for a variety of models, and expecting concept forgetting in recent foundation models is not surprising. However, we not only talk about concept forgetting, we propose a simple way to quantify it, analyze all *E2E fine-tuning* methods on concept forgetting, and use insights from this analysis to propose a new E2E fine-tuning method, LDIFS, that minimizes concept forgetting without compromising downstream task performance.
>
> *Expanding to Continual Learning Setups*
>
> * Due to the applicability of LDIFS to continual setups, we explore its effectiveness in such settings as well (Section 5, Table 2)  and also compare it with 5 other well-known continual learning baselines including LwF [c] and iCARL in Table 3 of the revised main paper. We find LDIFS to be consistently superior to all 5 CL baselines.
> * Finally, the simplicity of LDIFS also allows it to be scaled and applied to different architectures and foundation models like CLIP RN50 and FLAVA ViT-B/16 (Appendix C.7).
>
> Thus, our conceptual innovation compared to [a-c] lies in highlighting the importance of studying end-to-end fine-tuning methods from the lens of concept forgetting and bringing to light a simple and scalable fine-tuning approach that not only beats other fine-tuning methods but also outperforms well-known continual learning methods in minimizing forgetting.
>
>
> **"The proposed solution follows the knowledge distillation framework ... The innovation seems quite incremental."**
>
> * *Unlike classic distillation methods which distill either from logits or last layer features, LDIFS uses earlier features along the network in the distillation process as well* (details in Appendix B). This difference turns out to be crucial for LDIFS’s performance.
> * We empirically show this in Fig 11 (Appendix C.3 last paragraph) of the appendix where we compare with an ablation on LDIFS where we distill only from the last layer features. This ablation significantly underperforms compared to the version of LDIFS using the full feature vector.
> * As mentioned above, in Table 3 of the revised draft, we compare LDIFS with 5 distillation based continual learning baselines and find it to consistently outperform all these baselines on all the task sequences, thereby providing further empirical evidence on the importance of distilling from earlier features.

---

> > ### Author Response · Authors · 2023-11-20
> > **A Gentle Follow-up**
> >
> > Dear reviewer cA9V,
> >
> > Thank you again for the time you took to review our work!
> >
> > We believe that we have clarified your concerns regarding the conceptual innovation of our work and its improvements compared to distillation based CL.
> >
> > We would really appreciate it if you could have a look at our replies and let us know if you had further questions/comments.
> >
> > We hope to be able to effectively utilize the discussion phase in case you had any further questions or comments about our work.
> >
> > Thank you!

---

> > > ### Author Response · Authors · 2023-11-21
> > > **Utilizing the discussion phase**
> > >
> > > Dear reviewer cA9V,
> > >
> > > Thank you for your review on our work!
> > >
> > > We would appreciate it if you could please have a look at our replies and we are more than happy to get back to you if you have further questions or comments.
> > >
> > > We believe that this work holds value for the research community, as you and the other reviewers have also pointed out, and we want to make full use of the discussion phase to engage with you and clarify any further concerns that you may have.
> > >
> > > The author-reviewer discussion period ends tomorrow and so we will really appreciate if you could kindly have a look at our reply before that.
> > >
> > > Thank you very much again for your time and efforts on reviewing our paper!

---

> ### Comment · Reviewer_cA9V · 2023-11-22
>
> I appreciate the authors' effort in responding to my questions. However, I am not totally sure I am convinced by some of the responses.
>
> Focusing on E2E fine-tuning of foundation models is definitely interesting, yet I am not convinced that this is one of the conceptual innovations. Foundation models are deep model pre-trained on large-scale data. Prior works has pointed that pre-trained deep models show catastrophic forgetting when fine-tuned on downstream tasks, and this issue needs to be addressed. That is why various methods have been developed (such as iCaRL and LwF) to mitigate the forgetting. If we still consider foundation models are a type of pre-trained deep models, doing the same things does not seem to conceptually novel to me.
>
> I acknowledge that the paper presented a new E2E fine-tuning method. The question is that if the proposed method deems technically innovative. As mentioned in my previous comment, the proposed method is essentially feature-based knowledge distillation, which has been previously studied extensively. I shall refer the authors to Sec 2.2 in the survey paper [a] for many of the prior works that distill from intermediate features. Many of these feature-based methods has been considered for learning without forgetting (e.g., [b]). Again, I appreciate the empirical results yet have a hard time to consider this as a major contribution.
>
> Many previous methods to mitigate catastrophic forgetting also talked about continual learning. Yes, the empirical results are promising, yet expanding the proposed method to continual learning, as a conceptual innovation, seems a bit weak.
>
> [a] Gou, J., Yu, B., Maybank, S. J., & Tao, D. (2021). Knowledge distillation: A survey. International Journal of Computer Vision, 129, 1789-1819.
>
> [b] Jang, Y., Lee, H., Hwang, S. J., & Shin, J. (2019, May). Learning what and where to transfer. In International conference on machine learning (pp. 3030-3039). PMLR.
>
> Taking all these into consideration, I could not recommend this paper.

---

### Author Response · Authors · 2023-11-18
**General Response**

Thank you very much for taking the time to review our work! We appreciate your encouraging comments on the importance of the problem of concept forgetting (cA9V, VWXz, Deyx), the simplicity and performance of our proposed LDIFS regularizer (cA9V, VWXz, 2XG5, Deyx), our comprehensive empirical analyses (VWXz, 2XG5, Deyx) and the clarity and organization of the paper (cA9V, VWXz, 2XG5, Deyx).

**Contributions**
* We extensively analyse state-of-the-art end-to-end fine-tuning methods popularly used in foundation models and show that all these methods suffer from concept forgetting, a phenomenon where the model forgets pre-trained concepts as it is fine-tuned on a downstream task.
* We propose a simple method of quantifying concept forgetting in foundation models.
* Drawing insights from this analysis, we propose a new end-to-end fine-tuning method, which can significantly alleviate the problem of concept forgetting without compromising on downstream task performance.
* We conduct an extensive evaluation of our proposed fine-tuning method on both single downstream tasks as well as a sequence of downstream tasks and find it to perform excellently in all cases.

**Additional experiments and changes**: Based on your suggestions, we have made the following additions and changes to the paper.

***Main paper***

* *Introducing ImageNet*: We have included ImageNet as the 10th fine-tuning task in Table 1 of the paper. ImageNet is a medium sized classification dataset with 1000 classes. As seen in Table 1, even when fine-tuning on ImageNet, we find the phenomenon of concept forgetting to exist and also observe the superior performance of LDIFS.

* *Improved continual learning (CL) setup*: We have improved our continual learning setup. Instead of 2 sequences of 2 tasks each, we now have 3 sequences of 3 tasks each:  a) SVHN -> C10 -> RESISC45, b) SVHN -> C100 -> RESISC45, c) SVHN -> Cars -> RESISC45. We have updated Table 2, Figure 6 and Section 5 of the paper to reflect this change.

* *Additional comparisons with CL baselines*: In Table 3, we compare LDIFS on the continual learning setup with 5 knowledge-distillation based continual learning baselines: LwF [1], LFL [2], iCARL [3], D+R [4] and ZSCL [5].

* *Discussion on Distillation & Continual Learning*: We have expanded the Knowledge Distillation & Continual Learning paragraph in the Related Works section (Section 6) of the paper. We specifically highlight how LDIFS is different from other distillation based CL methods with results in Table 3 and Fig 11 of the Appendix to substantiate our claims on the importance of distilling from earlier features in the network.

***Appendix***:
* *Wise-FT*: In Appendix C.5 we provide an ablation combining fine-tuning baselines with Wise-FT. We find Wise-FT to improve performance on all baselines including LDIFS. The order of performance remains broadly the same with LDIFS being superior to others both pre and post Wise-FT.

* *Alternate visualization for Table 1*: In Appendix C.6 we provide an alternative visualization for Table 1. For each downstream task, we plot the LP accuracy on the task itself along the x-axis and the average LP accuracy on all other tasks on the y-axis. We find LDIFS to lie close to the top right corner of this plot indicating superior performance both from preventing concept forgetting as well as obtaining good performance on the downstream task at hand.

[1] Li, Z. and Hoiem, D., 2017. Learning without forgetting. IEEE transactions on pattern analysis and machine intelligence, 40(12), pp.2935-2947.

[2] Jung, H., Ju, J., Jung, M. and Kim, J., 2018, April. Less-forgetful learning for domain expansion in deep neural networks. In AAAI 2018.

[3] Rebuffi, S.A., Kolesnikov, A., Sperl, G. and Lampert, C.H., 2017. icarl: Incremental classifier and representation learning. In CVPR 2017

[4] Hou, S., Pan, X., Loy, C.C., Wang, Z. and Lin, D., 2018. Lifelong learning via progressive distillation and retrospection. In ECCV 2018.

[5] Zheng, Z., Ma, M., Wang, K., Qin, Z., Yue, X. and You, Y., 2023. Preventing Zero-Shot Transfer Degradation in Continual Learning of Vision-Language Models. ICCV 2023.